# Intensifying precipitation over the Southern Ocean challenges reanalysis-based climate estimates – Insights from Macquarie Island's 45-year record

Zhaoyang Kong[1,2], Andrew T. Prata[3], Peter T. May[1], Ariaan Purich[1,2], Yi Huang[4,5], Steven T. Siems[1,2]

[1]School of Earth, Atmosphere and Environment, Monash University, Melbourne, 3800, Australia

[2]Australian Research Council Special Research Initiative for Securing Antarctica's Environmental Future, Melbourne, 3800, Australia

[3]CSIRO Environment, Melbourne, 3168, Australia

[4]School of Geography, Earth and Atmospheric Sciences, The University of Melbourne, Melbourne, 3010, Australia

[5]Australian Research Council Centre of Excellence for the Weather of the 21st Century, Melbourne, 3010, Australia

*Correspondence to*: Zhaoyang Kong (zhaoyang.kong@monash.edu)

**Abstract.** The Southern Ocean (SO) region plays a critical role in the global climate system but remains observationally limited. Macquarie Island (MAC), situated along the SO storm track, provides a unique, high-quality surface precipitation record since 1948. Based on daily synoptic regime classifications from 1979 to 2023, we find that a significant 28% increase in annual precipitation at MAC is primarily driven by enhanced mean daily precipitation intensity associated with warm air advection, low pressure, and cold air advection regimes, rather than shifts in regime frequency, consistent with a poleward shift in the storm track. In contrast, ECMWF reanalysis (ERA5) shows only an 8% overall increase in annual precipitation, as it insufficiently reflects the increase in mean daily precipitation intensity under these regimes, likely due to its limited representation of atmospheric moisture transport and increasing evaporation. This precipitation discrepancy suggests that reanalysis may underestimate aspects of the moisture and energy budgets over the SO, and may also have potential implications for the estimation of SO freshwater fluxes.

## 1. Introduction

The broader Southern Ocean (SO) region, from the subtropical ridge to the Antarctic coast, is characterised by a nearly uninterrupted ocean expanse largely devoid of land masses and plays a vital role in the Earth's climate system (Rintoul and Church, 2022), taking up large portions of anthropogenic heat and carbon (Williams et al., 2024; Khatiwala et al., 2009; Gruber et al., 2023). The climate of this region is dominated by the SO storm track and its strong westerly winds, and is experiencing changes driven by global warming, and stratospheric ozone depletion and stabilisation (Cai et al., 2023). The westerly winds have strengthened and shifted poleward (Shaw et al., 2016; Yin, 2005) – reflecting a positive trend of the Southern Annular Mode (SAM) since the satellite era (post-1979) (Marshall, 2003; Dong et al., 2023) – and are associated with a redistribution of precipitation in reanalysis products and climate simulations, with the lower latitudes of the SO becoming drier and the higher latitudes experiencing greater precipitation (Manton et al., 2020).

Given that precipitation necessitates evaporation and a compensating latent heat flux (LHF), a primary means of cooling the SO mixed layer, precipitation serves as a constraint on the surface energy budget. In addition, precipitation, along with Antarctic glacial melt, is one of the primary sources of freshwater flux into the SO, modifying ocean buoyancy forcing, facilitating ocean mixing, impacting sea ice variability (Shaw et al., 2016; Li et al., 2000; Stammerjohn et al., 2011; Pauling et al., 2016), and having an impact on the biogeochemistry of the ocean mixed layer (Fisher et al., 2025). The estimation of precipitation over the SO is particularly challenging (Siems et al., 2022). The SO region has few continuous, in-situ observations due to its remoteness, lack of land masses, and harsh conditions along the storm tracks. The scarcity of in-situ observations not only restricts research on the weather and climate over the SO but also makes it challenging to evaluate numerical simulations and remote sensing products (McFarquhar et al., 2021; Gettelman et al., 2020; Hwang and Frierson, 2013; Bodas-Salcedo et al., 2019; Sallée et al., 2013; Lauer et al. 2023; Trenberth and Fasullo, 2010; Cesana et al., 2022). This ambiguity profoundly impacts the accuracy of atmospheric and oceanic circulation modelling and climate sensitivity studies, both across the SO and on a global scale (Zelinka et al., 2020; Kay et al., 2016).

Macquarie Island (Fig. 1) is an isolated island situated along the SO storm track, with a research station (MAC; 54.50° S, 158.94° E; elevation 8 m) located on a narrow isthmus at the northern end that minimises any orographic distortion (Wang et al., 2016). Its long-term, high-quality observations constitute a unique record for the SO that has been extensively used to evaluate satellite products, reanalysis data, and climate simulations (Wang et al., 2015; Tansey et al., 2022; Tansey et al., 2023). The surface record from MAC indicates that the island has been experiencing a statistically significant increase in annual precipitation over several decades (Adams, 2009), with an increase of 24% (~225 mm) from 1979 to 2021 (Siems et al., 2022). This increase is contributing to a degradation of the native flora of this United Nations Educational, Scientific and Cultural Organisation (UNESCO) World Heritage site (Bergstrom et al., 2015; Dickson et al., 2021). This additional precipitation is equivalent to a LHF of an additional 18 W m$^{-2}$ (see Methods), although it is unknown over what spatial extent either the precipitation or evaporation occurs. Comparing these surface observations to precipitation from the European Centre for Medium-Range Weather Forecasts Reanalysis v5 (ERA5), Siems et al. (2022) identified a bias, with the ERA5 precipitation showing only a 6% (~62 mm) increase in annual precipitation at MAC. This discrepancy, though previously documented, remains unexplained and forms the core motivation for the present work.

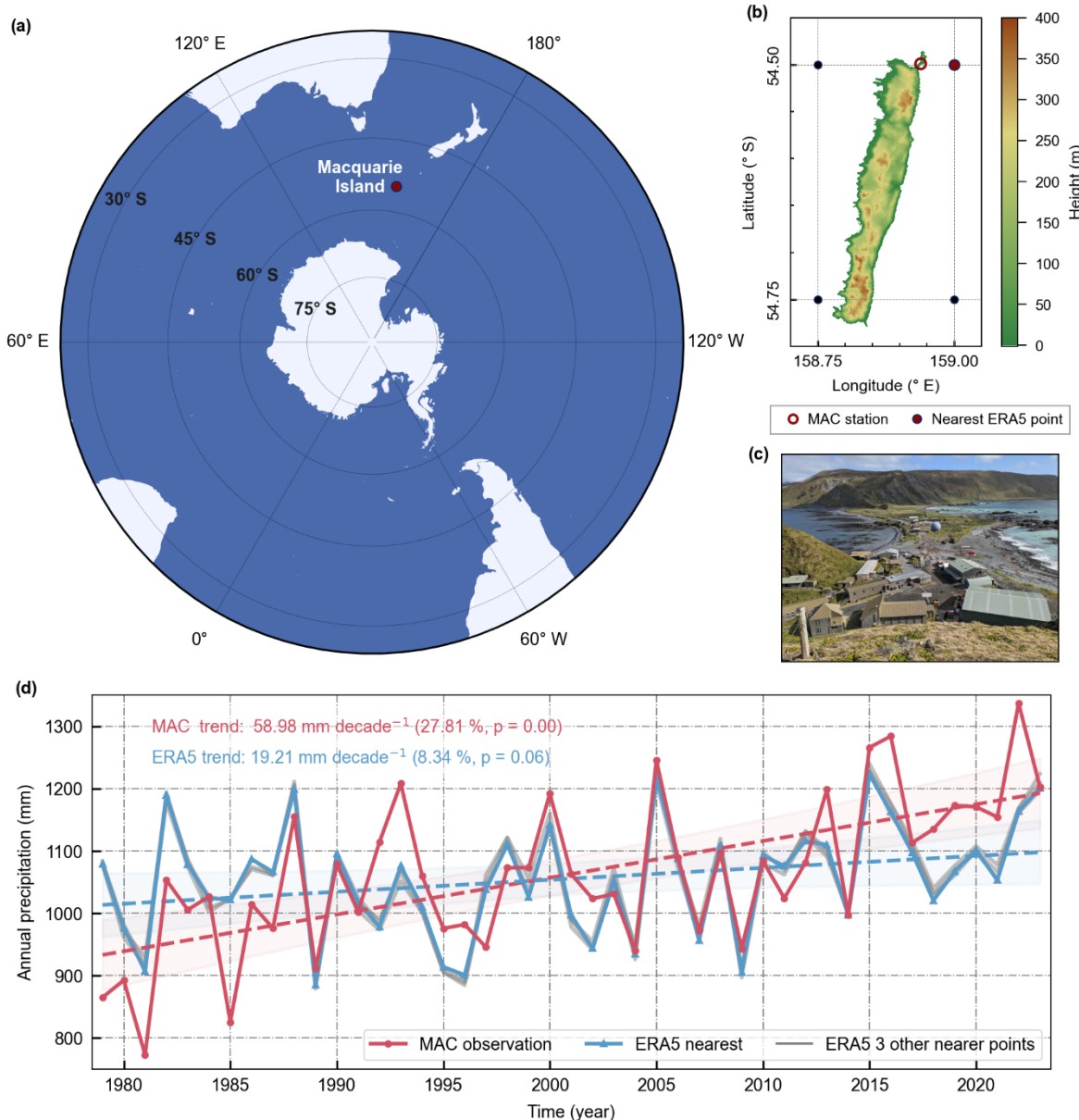

**Figure 1: (a)** The location of Macquarie Island (MAC, red solid dot) in the Southern Ocean. **(b)** The topographic elevation map of MAC, which is approximately 34 km long and 5.5 km wide, with a highest point around 410 m. The MAC station is located at the isthmus at the northern end of the island (white dot with red border) at an elevation of 8 m. The nearest ERA5 grid point to the MAC station (red solid dot) is located approximately 4 km east over the ocean, along with three other ERA5 grid points at a 0.25° resolution near the island (dark blue dots). **(c)** Photograph of the MAC station site (view from the north). © Photo: Brendan Barnes / Australian Antarctic Division (AAD) (2020) (AAD, 2021). **(d)** Comparison of annual precipitation trends from 1979 to 2023 between MAC observations (red), nearest ERA5 grid point (blue), and three other nearby ERA5 grid points (grey) shown in panel (b), with shaded areas indicating the 95% confidence intervals.

This study aims to investigate the meteorology behind the observed increase in annual precipitation at MAC through the following objectives: (1) Classify daily synoptic meteorological patterns over the past 45 years (1979-2023). (2) Examine the frequency, intensity (mean daily precipitation), and contributions of different synoptic regimes to the observed cumulative precipitation at MAC. (3) Compare MAC observational data with ERA5 output at the nearest grid point to evaluate biases in precipitation trends across all classifications. (4) Investigate how changes in synoptic meteorology reflect intensifying climatic trends in precipitation at MAC, address

discrepancies in ERA5, and discuss the broader implications.

## 2. Data and Methods

### 2.1 Data

#### 2.1.1 Macquarie Island Observation Data

The Macquarie Island station (MAC, BoM station ID. 300004) is operated by the Australian Antarctic Division (AAD) with support from the Australian Bureau of Meteorology (BoM). The daily precipitation accumulation records are measured over a 24-hour period from 09:00 to 09:00 local time (LT). This daily update in Local Standard Time (LST, UTC + 10) corresponds to 23:00 UTC, while during Daylight Savings Time (DST, UTC + 11) it corresponds to 22:00 UTC. Only 7 days are missing from the MAC daily precipitation dataset from 1979 to 2023, ensuring data continuity for the analysis. The records are primarily collected using tipping bucket rain gauges that generally have a resolution of 0.2 mm, although measurements with a 0.1 mm resolution were recorded on 18 days during this period. The observed mean sea level pressure (MSLP) and 2-m temperature data from MAC are from the BoM Climate Data Services (CDS). Some data gaps are filled by data from the U.S. National Centres for Environmental Information (NCEI) Global Hourly Integrated Surface Database (ISD) at the MAC site (World Meteorological Organisation (WMO) station ID. 94998).

#### 2.1.2 ERA5 Reanalysis Data

The European Centre for Medium-Range Weather Forecasts (ECMWF) Reanalysis v5 (ERA5) (Hersbach et al., 2020) is the fifth-generation global weather and climate reanalysis dataset. In this study, we used the ERA5 pressure level data with 37 vertical levels, a horizontal resolution of 0.25° × 0.25°, and an hourly temporal resolution. The nearest ERA5 grid point to MAC is located at 54.50° S, 159.00° E, approximately 4 km east of the MAC observation station over the ocean. When converting ERA5 hourly data into daily values, we disregarded the effects of DST and processed the data based on MAC LST (UTC + 10), namely using 23:00 UTC as the start of the daily interval, to ensure alignment with the observed precipitation records.

#### 2.1.3 Macquarie Island Topographic Data

The digital elevation model (DEM) data for Macquarie Island (Fig. 1b) is from the Shuttle Radar Topography Mission Global 1 arc second (SRTMGL1) dataset (Farr et al., 2007; NASA JPL, 2013), led by the U.S. National Geospatial-Intelligence Agency (NGA) and National Aeronautics and Space Administration (NASA). It provides surface elevation information with a resolution of 1 arc second (approximately 30 meters).

### 2.2 Methods

#### 2.2.1 Daily Synoptic Systems Classification

We employed the K-means clustering method based on the Scikit-Learn library of Python (Pedregosa et al., 2011). For the synoptic typing using data from the nearest ERA5 grid point at 11:00 UTC (21:00 LST), 15 standardised variables, including air temperature, relative humidity, zonal wind, and meridional wind at three pressure levels (925, 850, and 700 hPa) in the lower troposphere, and three surface variables (surface pressure, 2-m air

temperature, and relative humidity) were used following Lang et al. (2018) and Truong et al. (2020). The value of K (i.e. number of clusters) needs to be specified in advance. Once chosen, the K-means algorithm generates K cluster centroids (mean values in each cluster).

In order to decide upon an appropriate number of clusters, we tested K values ranging from 3 to 8 and finally determined that K = 5 was optimal for classifying daily synoptic patterns at MAC from 1979 to 2023, categorised as zonal flow, warm air advection, low pressure, cold air advection, and high pressure. These cluster types are broadly consistent with those identified in earlier studies that were based on shorter timeframes (Lang et al., 2018; Truong et al., 2020).

### 2.2.2 48-hour Atmospheric Back Trajectory

The Hybrid Single-Particle Lagrangian Integrated Trajectory (HYSPLIT) model (Stein et al., 2015), developed by the U.S. National Oceanic and Atmospheric Administration (NOAA), is widely used for forward and backward tracking of air mass trajectories. We used the HYSPLIT model driven by ERA5 single-level and pressure-level data to compute 48-hour back trajectories starting at 11:00 UTC each day from January 1st 1979 to December 31st 2023. The trajectories were initialised at an altitude of 500 m, with the starting point set at the nearest ERA5 grid point to Macquarie Island (54.50° S, 159.00° E).

To convert ERA5 data from GRIB format to the ARL format required for HYSPLIT, we used the MeteoInfo software (Wang, 2014).

### 2.2.3 Relationship of Annual Precipitation, Frequency Proportion, and Daily Intensity

For each cluster, we have Eq. (1):

$$P_{i,j} = f_{i,j} \times \bar{p}_{i,j} \times D_j , \tag{1}$$

where $P_{i,j}$ represents accumulated precipitation, $f_{i,j}$ represents frequency proportion in percentage, $\bar{p}_{i,j}$ represents mean daily precipitation, $D_j$ represents days per year, could be 365 or 366 days, $i$ represents clusters, from 1 to 5, $j$ represents year, from 1979 to 2023. Note: This formula can also be applied to the overall precipitation, where the yearly frequency proportions are always 100%.

Since the number of days in a year is nearly constant (365 or 366 days), and the total sample has only 7 missing days, we can consider that the annual accumulated precipitation for each cluster is determined by the frequency proportion of cluster in that year and the average daily precipitation intensity, with these two variables having a product relationship.

### 2.2.4 Regression Analysis and Trend Significance Verification

In this study, we employed a linear regression model based on the Ordinary Least Squares (OLS) method to analyse the trends (Seabold and Perktold, 2010). To assess the significance of the regression line, we calculated the p-value of the slope coefficient. A p-value less than 0.05 was considered indicative of a statistically significant trend. Furthermore, we computed the 95% confidence interval for the regression line to evaluate the reliability of the results. These analytical approaches were employed to ensure the robustness and accuracy of the trend analysis.

**2.2.5 Relationship between Latent Heat Flux and Evaporation in ERA5**

Assuming the water density is $\rho = 10^3 \, kg \, m^{-3}$. The latent heat of vaporization of water at 0 °C is $L_v = 2.5 \times 10^6 \, J \, kg^{-1}$ ([https://glossary.ametsoc.org/wiki/Latent_heat](https://glossary.ametsoc.org/wiki/Latent_heat)). Therefore, on a daily scale, the factor between latent heat flux (LHF) and evaporation can be expressed as Eq. (2):

$$\frac{LHF}{evaporation} = \frac{L_v}{86,400 \, s} \approx 28.94 \, J \, kg^{-1} \, (s \, day^{-1})^{-1} \, , \tag{2}$$

which can be confirmed in Fig. 4b-middle at the MAC site.

Assuming that in the ERA5 model, long-term precipitation equals evaporation, i.e., the precipitation variable is equal to the evaporation variable. From 1979 to 2021, MAC observed an increase of 225 mm in precipitation, the corresponding LHF is calculated as Eq. (3):

$$LHF = \frac{225 \, mm}{365.25 \, day} \times 10^3 kg \, m^{-3} \times 28.94 \, J \, kg^{-1}(s \, day^{-1})^{-1} \approx 18 \, W \, m^{-2} \, , \tag{3}$$

similarly, for the linear regression bias between ERA5 and MAC observation in 2023 (predicted) equals 95 mm (Fig. 1d), corresponding to an LHF of approximately 7.5 W m-2.

**3. Results**

**3.1 Synoptic Typing at Macquarie Island**

Five distinct regimes were found to be optimal for characterising the progression of mid-latitude cyclones and associated fronts along the SO storm track.

- Zonal flow (ZFL, Fig. 2a-c) is characterised by strong westerly winds with little vertical shear, a relatively weak inversion at 900 hPa and moist air. A strong zonal gradient is evident in mean sea level pressure (MSLP), and the 48-hour boundary layer HYSPLIT back trajectories are mainly limited to the 50-60° S zonal band.
- Warm air advection (WAA, Fig. 2d-f) is characterised by the warm conveyor belt ahead of a mid-latitude cold front with strong winds (~10.2 ms-1) from the northwest, a high pressure system northeast near New Zealand, no clear boundary layer inversion and warm, moist air aloft. 48-hour back trajectories primarily originate from lower latitudes (40-55° S) with an anti-cyclonic rotation.
- Low pressure (LPR, Fig. 2g-i) is characterised by weak surface winds with no boundary layer inversion, but strong vertical shear aloft. Above the boundary layer, the air is very moist with a dew point depression of just 6.0 °C at 700 hPa. A strong cyclonic rotation is evident in the MSLP and 48-hour back trajectories.
- Cold air advection (CAA, Fig. 2j-l) is characterised by post-frontal air masses passing MAC. Winds are southwesterly with a relatively strong inversion evident at 850 hPa with cold and dry air aloft. There is a cyclonic rotation off the coast of Antarctica near 180° E, and the back-trajectories originate at higher latitudes with many reaching the coast of Antarctica.
- High pressure (HPR, Fig. 2m-o) is characterised by ridging over MAC, with weak westerly winds. The boundary layer inversion is evident at 950 hPa with a nearly isothermal layer above extending to 850 hPa. Above the inversion, the air is warm and dry. The 48-hour back trajectories are shorter in length and show a considerable spread across the SO.

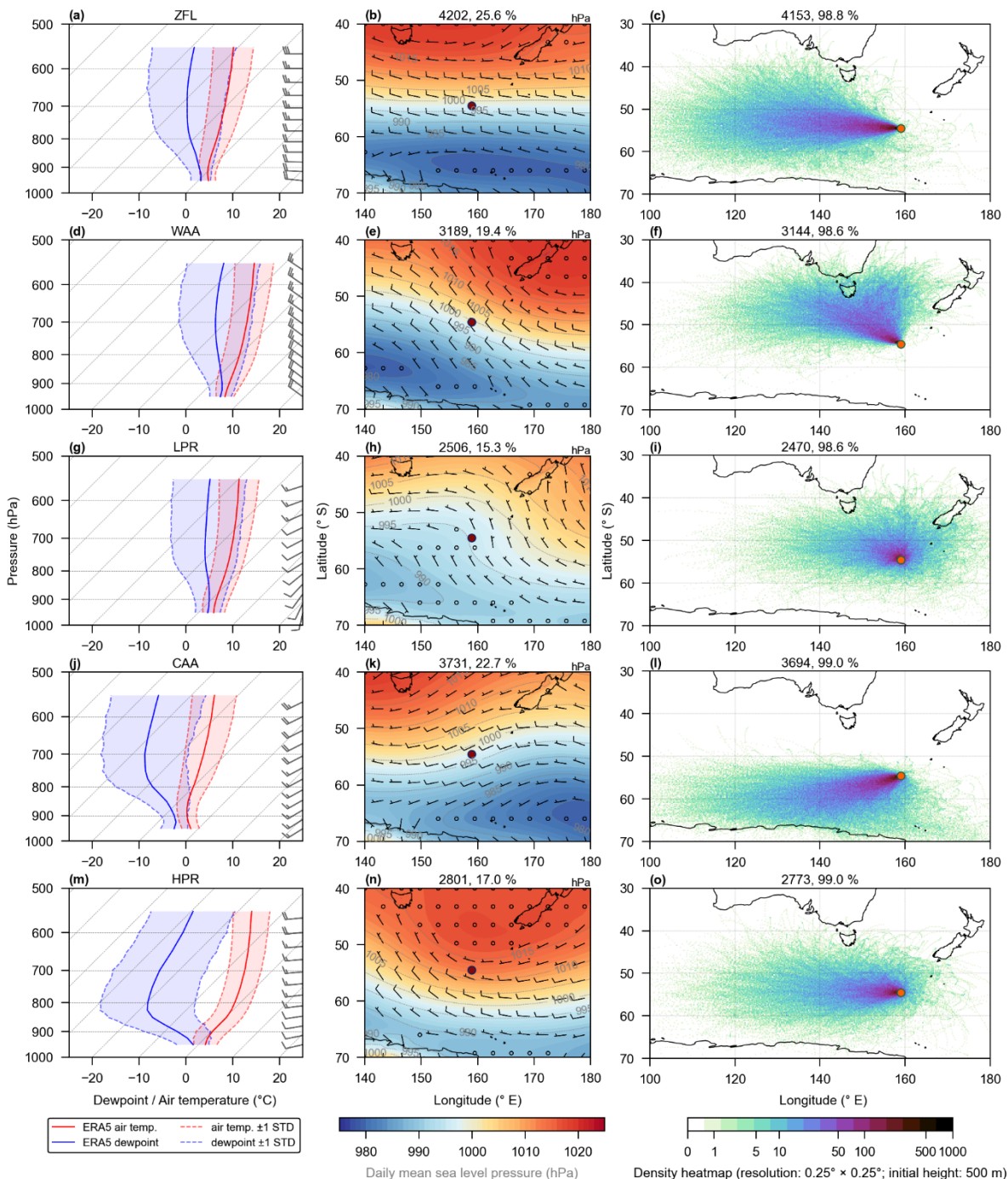

**Figure 2: For each synoptic type of k-means clustering from 1979 to 2023 based on ERA5 datasets. Zonal flow (ZFL),**
**Warm air advection (WAA), Low pressure (LPR), Cold air advection (CAA), and High pressure (HPR). Column 1 (a,**
**d, g, j, m) The average air temperature (°C, solid red) and dewpoint (°C, solid blue) profiles from 950-550 hPa at 11**
**UTC (around 21 Local Time) with ±1 standard deviations (dashed lines), and wind vector profiles (black). Column 2**
**(b, e, h, k, n) The mean daily mean sea-level pressure (MSLP, hPa, colour-filled and grey lines) and 10-m wind vector**
**(m s⁻¹, black wind barb) around Macquarie Island (MAC, red point). Column 3 (c, f, i, l, o) The 48-hour back**
**trajectories density heatmaps with the resolution of 0.25° × 0.25°, initialized at 500 m from MAC (orange point).**

Given that primary MAC observations are assimilated into ERA5, a good agreement in MSLP and 2-m air
temperature is observed across the synoptic types (Fig. 3a). The daily mean precipitation values are also

comparable (Fig. 3a), although ERA5 precipitation is derived solely from model output and does not directly
assimilate MAC observations. The low pressure and warm air advection regimes exhibit the highest daily
precipitation rates (4.3 and 4.2 mm day$^{-1}$, respectively), whereas the high pressure regime has the lowest daily
precipitation rate (0.9 mm day$^{-1}$). From the first decade (1979-1988) to the last (2014-2023), the overall daily
mean precipitation at MAC increases 0.6 mm day$^{-1}$, primarily during warm air advection, cold air advection, and
low pressure regimes (Fig. 3b). In contrast, ERA5 only records an increase of 0.2 mm day$^{-1}$, primarily driven by
the low pressure regime.

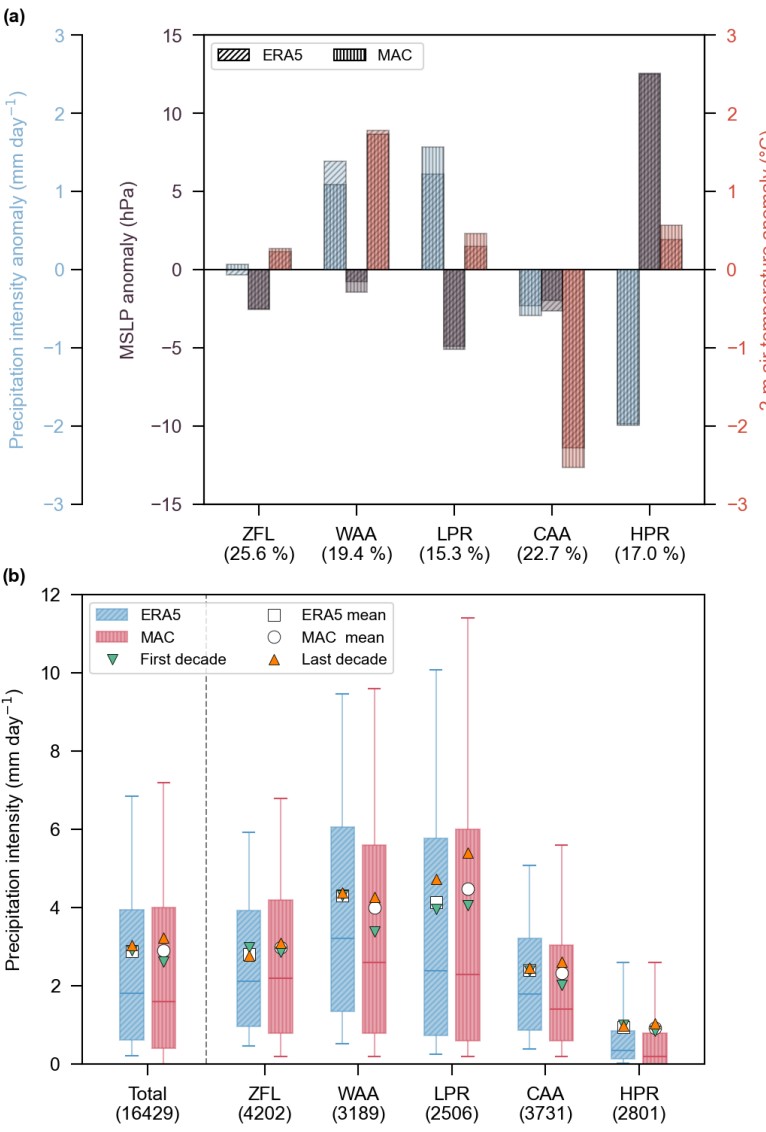

**Figure 3: Mean surface values for each regime based on both of the MAC observations (vertical lines) and the nearest
ERA5 (diagonal lines) datasets. (a) Surface daily anomalies of precipitation intensity (mm day$^{-1}$, blue), mean sea-level
pressure (MSLP, hPa, grey) and 2-m air temperature (°C, red). (b) Box and whiskers plot for overall (total) and each
regime of daily precipitation intensity in the percentile range of 10th and 90th with mean values (square for MAC,
circle for ERA5), as well as the means of the first decade (1979-1988, green lower triangle) and the last decade (2014-
2023, orange upper triangle).**

A strong seasonality is observed for these five synoptic regimes along with the key physical measurements of the

environment (Fig. 4). The monthly averaged sea surface temperature (SST) reaches a peak in February (6.9 °C) and a minimum in August (4.2 °C) reflecting the thermal lag in the remote ocean SST (Fig. 4b-bottom). SST variations have a strong impact on the synoptic meteorology, being highly correlated with the occurrence of warm air advection (0.96), cold air advection (-0.97) and high pressure (0.88) regimes. The warm air advection and high pressure regimes are most common during a lagged austral summer (Jan-Feb-Mar) at 27% and 21%, respectively, while cold air advection is most common during a lagged austral winter (Jul-Aug-Sep) at 32% (Fig. 4a). This seasonal cycle of the synoptic meteorology aligns with the seasonal cycle of the SO storm track: during the summer the storm track advances poleward with warm air advection and high pressure regimes more common over MAC, while during the winter the storm track retreats to lower latitudes and cold air advection is the dominant synoptic pattern over MAC (Hoskins and Hodges, 2005).

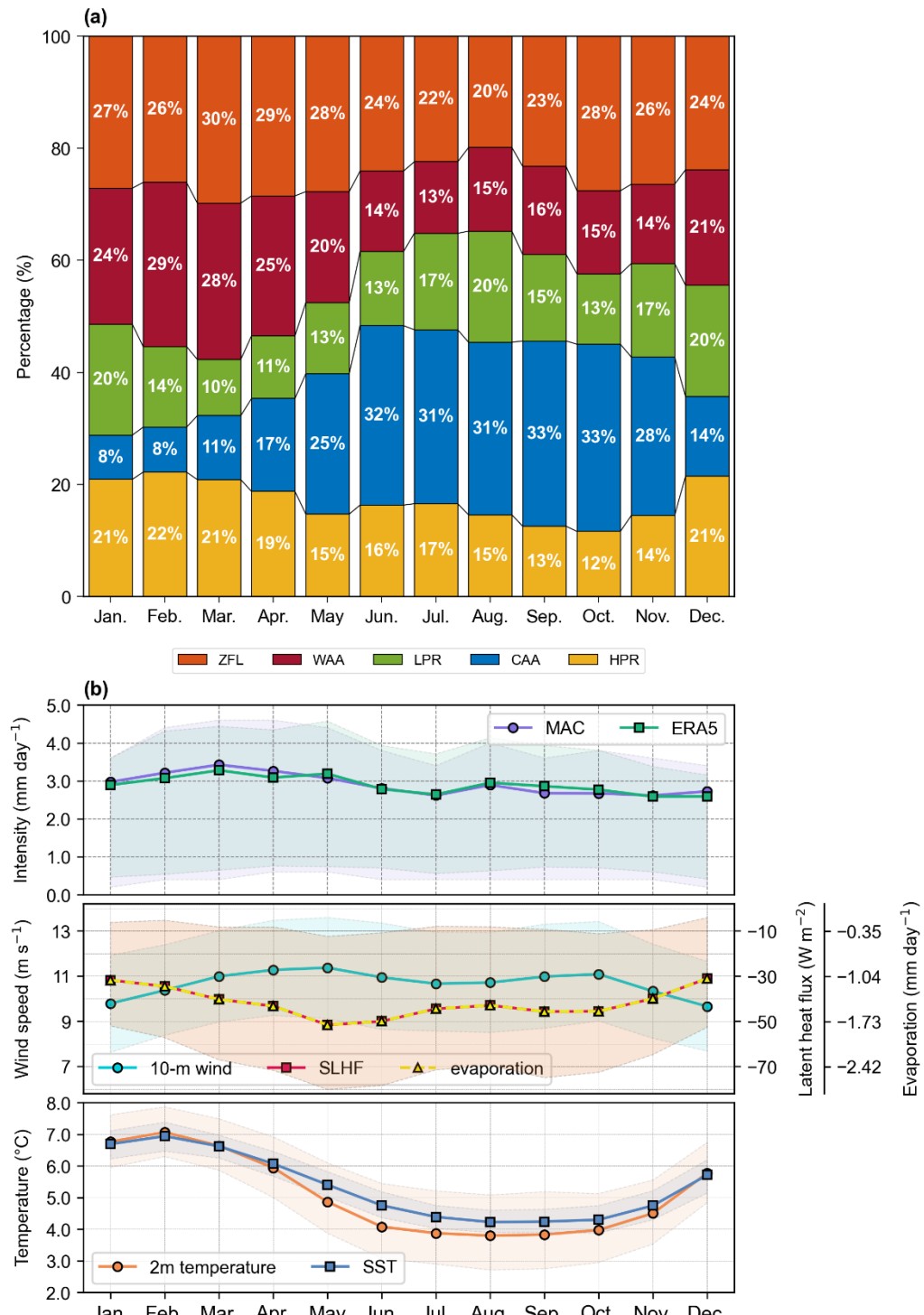

**Figure 4: (a) Mean monthly frequency of occurrence (%) of each cluster from 1979 to 2023. (b) Monthly means based on daily variables. Precipitation intensity (mm day⁻¹) for MAC observations (purple) and ERA5 (green) (top). ERA5 10-meter wind speed (m s⁻¹, green), surface latent heat flux (SLHF, W m⁻², red), and evaporation (mm day⁻¹, yellow) (middle). ERA5 2-meter air temperature (°C, orange) and SST (°C, blue) (bottom). The shaded areas for each variable represent the range from the 25th to the 75th percentile.**

The seasonal cycle of the two remaining synoptic patterns, zonal flow and low pressure, is negatively correlated with each other (-0.68), with the zonal flow peaking in March and reaching a minimum in August. The low pressure regime is most strongly correlated with the ERA5 10-m wind speed (-0.78), which peaks twice through

the year in May and October and is negatively correlated with the surface LHF/evaporation (Fig. 4b-middle, and see Methods), reflecting the dependence of heat fluxes on the surface wind speed. Following the ERA5 convention, upward fluxes are negative indicating a cooling of the ocean and warming of the atmosphere. MAC precipitation is relatively steady through the year with a weak peak in March (3.4 mm day$^{-1}$) and a minimum in July and November (2.6 mm day$^{-1}$; Fig. 4b-top). Annually averaged, both the MAC and ERA5 precipitation (2.9 mm day$^{-1}$) far exceed the ERA5 evaporation (1.4 mm day$^{-1}$) indicating that moisture is converging towards the storm track through some combination of warm air advection from the north and cold air advection from the south.

## 3.2 Trends in the Precipitation

We now employ these synoptic patterns to explore the bias in the ERA5 precipitation trend. Starting with the frequency of the different regimes, only the warm air advection regime has a statistically significant trend in frequency of occurrence over the 45 years (1979-2023), increasing from 63.2 days per year in 1979 to 78.5 days in 2023 (Table 1). Over the same period, low pressure days decrease from 60.9 to 50.5 days, followed by cold air advection days, which decrease from 85.4 to 80.4 days. However, neither of these decreases is statistically significant (p = 0.08 and 0.41, respectively), due to the large year-to-year variability in the frequency of regimes.

| Annual occurrence frequency by clusters, 1979-2023 | | | | | |
|---|---|---|---|---|---|
| | 1979-predicted (day) | 2023-predicted (day) | change (day) | change (%) | p-value |
| ZFL | 92.6 | 94.2 | 1.6 | 1.8 | 0.76 |
| WAA | 63.2 | 78.5 | 15.3 | 24.2 | **0.00** |
| LPR | 60.9 | 50.5 | -10.4 | -17.0 | 0.08 |
| CAA | 85.4 | 80.4 | -5.0 | -5.9 | 0.41 |
| HPR | 63.3 | 61.1 | -2.2 | -3.5 | 0.71 |

**Table 1: Trends of annual occurrence by number of days for each cluster with increase change and slope p-value (p < 0.05 indicating a statistically significant trend, shown in bold). Note: The values for 1979 (predicted) and 2023 (predicted) represent the endpoints of the corresponding regression lines.**

Turning to the 45-year trends in the annual mean daily precipitation (Fig. 5), the intensity of the daily MAC precipitation increases 28.1% (2.6 to 3.3 mm day$^{-1}$), while ERA5 only increases 8.5% (2.8 to 3.0 mm day$^{-1}$), both of which are statistically significant. Breaking this down into the five synoptic patterns, MAC has a positive trend for each one, gaining statistical significance for the warm air advection, low pressure and cold air advection regimes. For ERA5, only the increase in the daily intensity of the low pressure regime is statistically significant. In contrast, the trends for warm and cold air advection show slight increases, while the zonal flow and high pressure regimes exhibit minor negative trends; however, none of these are statistically significant.

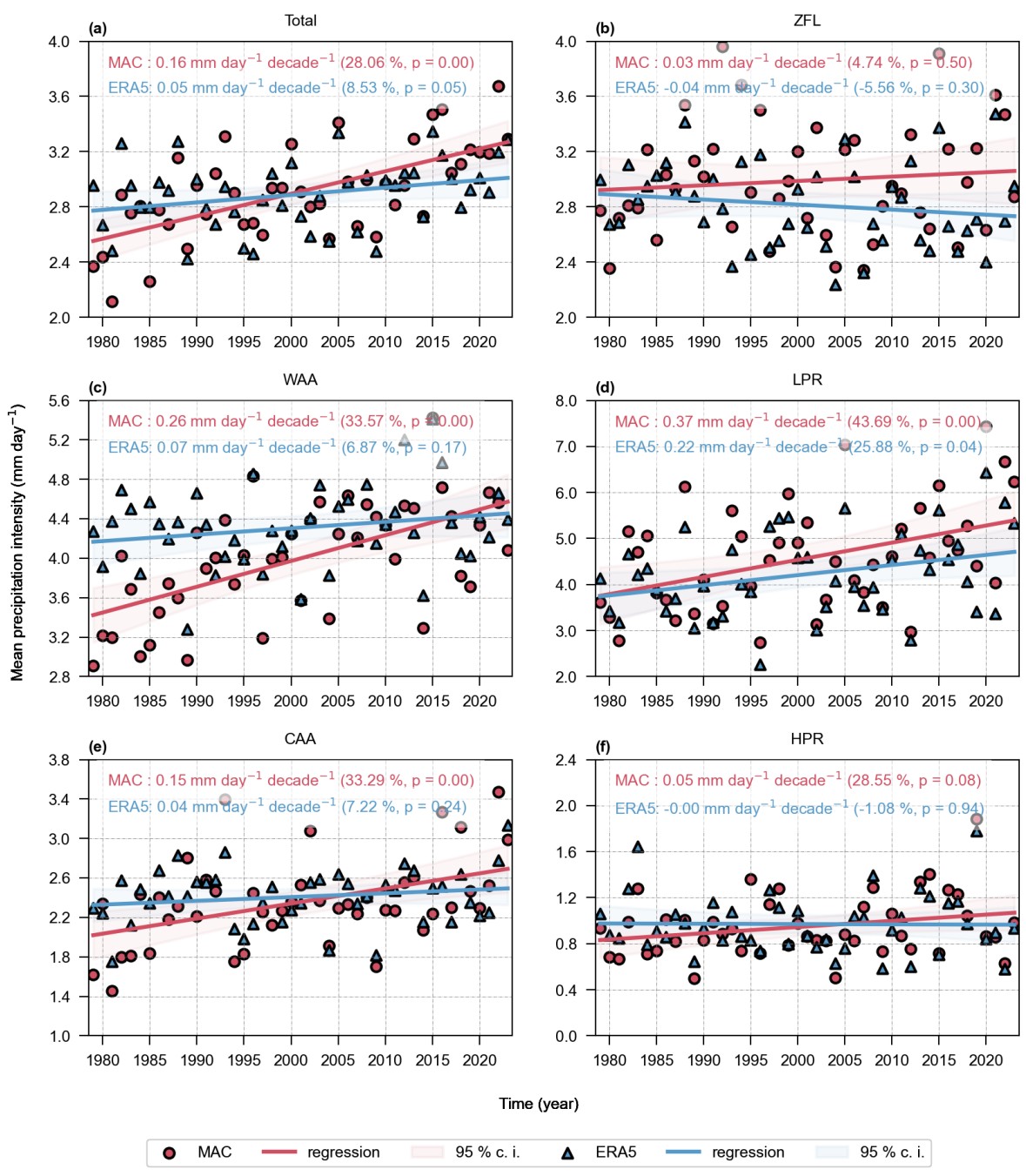

**Figure 5: Comparison of trends in annual mean daily precipitation intensity (mm day⁻¹) from 1979 to 2023 between MAC observations (red circle) and the nearest ERA5 (blue triangle), shown both overall (a) and separated by individual clusters (b-f). The captions on each panel report the 10-year trend in daily precipitation intensity (mm day⁻¹ 10 years⁻¹), the percentage of increase (%), and the slope p-value (with p < 0.05 indicating a statistically significant trend).**

Finally, the trends in the annual accumulated precipitation (Fig. 6) are considered. Overall, the least squares fit in the annual accumulated MAC precipitation increases by 260 mm (from 933 to 1193 mm or 27.8%) over the 45 years, while the accumulated ERA5 precipitation estimates only increase by 85 mm (from 1013 to 1098 mm or 8.3%). The bias is evident throughout the year, being slightly stronger in autumn (Apr-May-Jun) and winter (Jul-Aug-Sep) (Table 2).

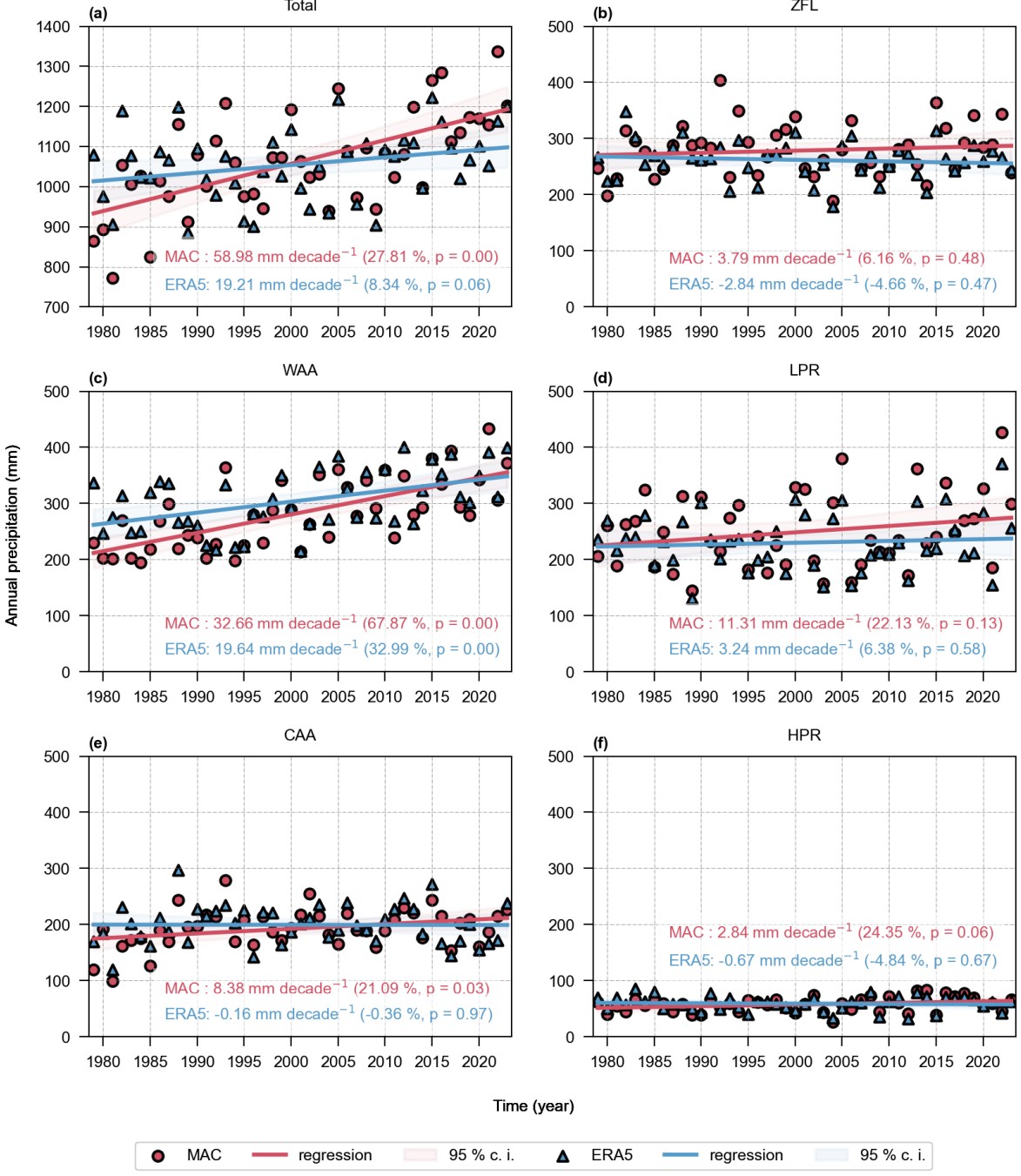

**Figure 6: Same as Figure 5 but for annual precipitation (mm).**

| Seasonal accumulated precipitation, 1979-2023, MAC obs. and ERA5 | | | | | |
|---|---|---|---|---|---|
| | 1979-predicted (mm) | 2023-predicted (mm) | change (mm) | change (%) | p-value |
| MAC | 263 | 315 | 52 | 19.7 | **0.02** |

| | | | | | | |
|---|---|---|---|---|---|---|
| Summer (Jan-Feb-Mar) | ERA5 | 272 | 284 | 12 | 4.7 | 0.54 |
| Autumn (Apr-May-Jun) | MAC | 238 | 317 | 79 | 33.2 | **0.01** |
| | ERA5 | 263 | 286 | 23 | 8.8 | 0.34 |
| Winter (Jul-Aug-Sep) | MAC | 215 | 287 | 72 | 33.6 | **0.00** |
| | ERA5 | 248 | 270 | 22 | 9.1 | 0.29 |
| Spring (Oct-Nov-Dec) | MAC | 217 | 274 | 57 | 26.1 | **0.01** |
| | ERA5 | 231 | 257 | 26 | 11.3 | 0.19 |

**Table 2: Seasonal accumulated precipitation changes (mm) for MAC and nearest ERA5 from 1979 to 2023, with the season definition including a 1-month lag.**

Assuming that changes in mean daily intensity are independent of the changes in the frequency of occurrence, we can linearly decompose the increase in accumulated precipitation by regime (Fig. 7). First, we use the observed changes in the frequency of occurrence (Fig. 7a) while fixing the mean daily intensity to find that the increase in the occurrence of the warm air advection regime (15.3 days) drives an increase of 62 mm in the annual accumulated precipitation. The corresponding decreases in the frequency of occurrence of the low pressure and cold air advection regimes, however, lead to decreases of 47 and 11 mm, respectively, over the 45 years. Aggregating the contributions across all five regimes, changes in the synoptic meteorology account for only 7 mm of 260 mm increase in annual MAC precipitation from 1979 to 2023. For ERA5, the changes in the synoptic meteorology account for only 14 mm of the 85 mm increase in accumulated precipitation.

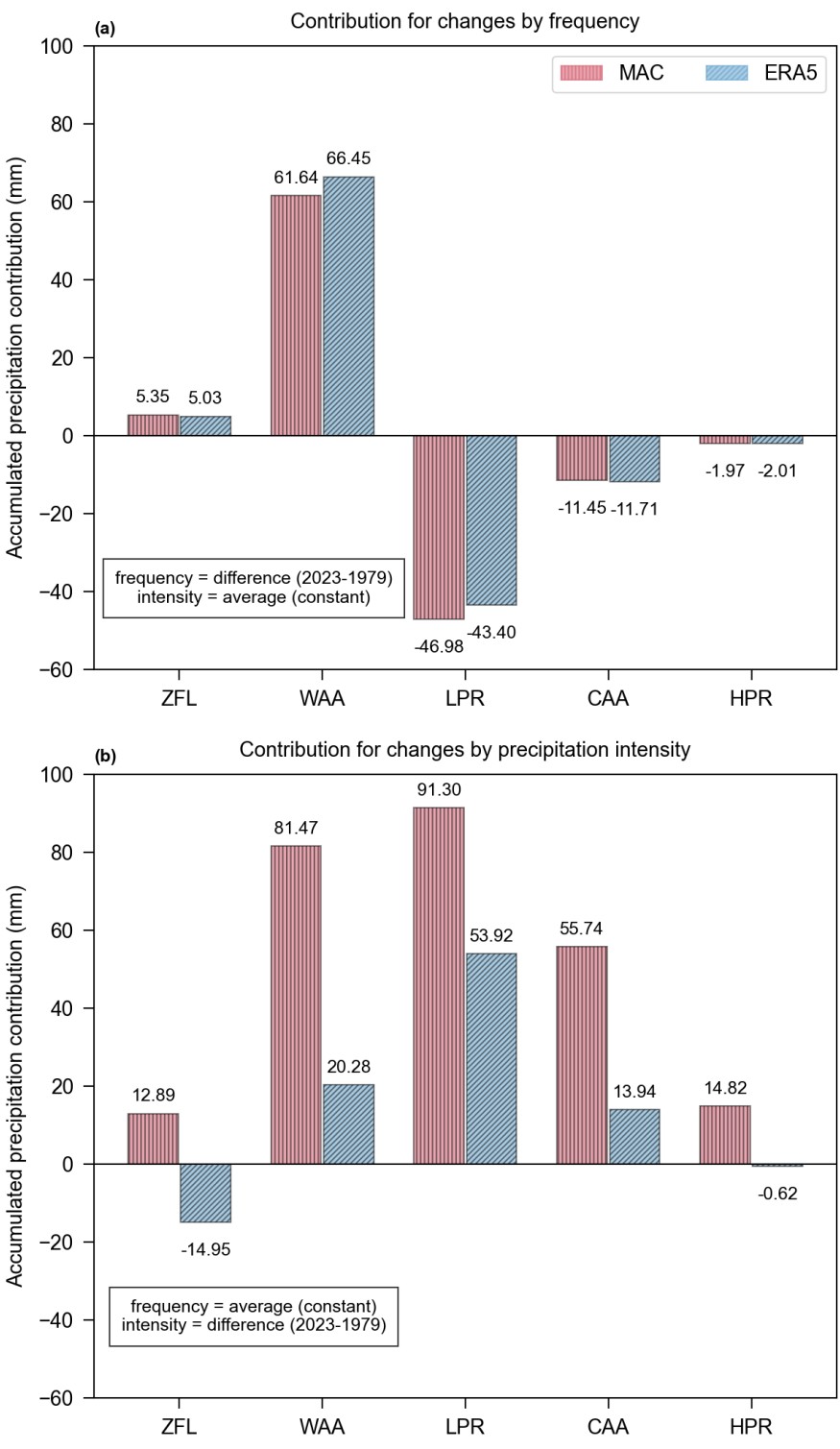

**Figure 7: Estimations of total contribution by each cluster to the increase in annual precipitation (mm) from 1979 to 2023 based on MAC observations (red vertical lines) and nearest ERA5 (blue diagonal lines) datasets, respectively. The contribution of changes by the occurrence frequency (a) and by the mean daily precipitation intensity (b) are derived from the Ordinary Least Squares (OLS) regression.**

305

We next examine the contribution from changes in the mean daily intensity when the average annual frequency of occurrence is held constant (Fig. 7b). Summed across all regimes, the increase in the daily intensity accounts for an increase of 256 mm in the MAC record, being led by increases in the low pressure (91 mm), warm air

advection (82 mm) and cold air advection (56 mm) regimes. For ERA5, the changes in the daily intensities account for an overall increase of 73 mm, coming primarily from the low pressure regime (54 mm).

## 4. Discussion and Conclusions

While MAC is only a single point along the SO storm track, the record is invaluable in our efforts to understand climate change across the Southern Ocean. The site has experienced a considerable and statistically significant 28% increase (260 mm) in annual precipitation over the 45-year record spanning from 1979 to 2023, while ERA5 is found to have a substantial bias, recording only an 8% increase (85 mm) (Fig. 1d). Our five synoptic regimes display a clear seasonality associated with seasonal variations in surface conditions and the storm track. Over these 45 years, the warm air advection regime has become more common over this period, increasing by ~15 days, while the low pressure and cold air advection regimes have decreased by ~10 and ~5 days, respectively. These changes reflect both a potential intensification (Reboita et al., 2015; Chemke et al., 2022), especially at higher latitudes, and a poleward shift of the storm track (Yin, 2005) – features linked to complex atmospheric and oceanic changes across the broader SO region (Shaw and Stevens, 2025), under the background of combined greenhouse gas increases and stratospheric ozone depletion (Swart et al., 2018).

Decomposing this increase in accumulated precipitation into the components arising from changes in frequency of occurrence and changes in mean daily intensity, we find that the shift in the storm track has a small impact on the overall increase in accumulated precipitation for both MAC and ERA5 (Fig. 7a). This is understandable as one heavy precipitation system (low pressure) is largely being replaced by another (warm air advection). ERA5 captures these trends in the synoptic meteorology, as in addition to the extensive use of satellite data, MAC soundings are also assimilated into the reanalysis and are heavily weighted given the remoteness of the location. This result suggests that it is the increase in the mean daily intensity of the different synoptic patterns that drives the increase in accumulated MAC precipitation. Large positive trends in mean daily intensity are evident for warm air advection, low pressure and cold air advection regimes, suggesting that moisture is converging towards the storm track from both low and high latitudes. Combined, the increase in intensity for these three systems accounts for 229 mm of the 260 mm increase in annual precipitation over the 45 years (Fig. 7). The increases in mean daily intensity for these three systems are central to the bias in ERA5, accounting for an annual increase of only 88 mm.

Although more supporting evidence is needed, assuming this increase in annual precipitation at MAC is representative of changes across the full SO storm track, it could have implications for the water and energy budgets over the broader SO. As water in the atmosphere is conserved, the 95 mm underestimation of the ERA5 2023 annual precipitation alone (Fig. 1d) requires either a corresponding overestimation of precipitation elsewhere or an underestimation in evaporation. One potential source of water into the broader SO atmosphere is through a flux from the subtropics, for example from atmospheric rivers (Reid et al., 2020; Rauber et al., 2020), which have been found to increase precipitation widely across the mid-latitudes (Finlon et al., 2020).

Alternatively, an underestimation of evaporation across the broader SO could produce biases in both the warm and cold air advection into the storm track, as observed, being driven by some combination of increasing westerly winds at higher latitudes and ocean surface warming north of ~40° S (Swart et al., 2018). Such a bias in

evaporation would also contribute to the bias in the energy budget over the SO (Trenberth and Fasullo, 2010; Schuddeboom and McDonald, 2021; Hyder et al., 2018), given that 95 mm year$^{-1}$ of precipitation corresponds to 7.5 W m$^{-2}$ LHF (see Methods). Within ERA5, LHF/evaporation is governed by a bulk parameterisation, being sensitive to both surface winds and the saturation humidity at the sea surface and the specific humidity of the near-surface air. Within ERA5, the annually averaged 10-m wind speed at MAC has increased 4.3% (10.5 to 10.9 m s$^{-1}$, Table 3) from 1979 to 2023, yet the annually averaged evaporation has had no significant change (Table 4).

| Annual mean daily wind magnitude changes by clusters, 1979-2023, ERA5 only | | | | | |
|---|---|---|---|---|---|
| | 1979-predicted (m s$^{-1}$) | 2023-predicted (m s$^{-1}$) | change (cm s$^{-1}$ decade$^{-1}$) | change (%) | p-value |
| Total | 10.5 | 10.9 | 10.2 | 4.3 | **0.02** |
| ZFL | 12.5 | 12.5 | 0.0 | 0.0 | 0.99 |
| WAA | 11.5 | 11.9 | 9.4 | 3.6 | **0.02** |
| LPR | 7.8 | 8.1 | 7.6 | 4.3 | 0.13 |
| CAA | 11.0 | 11.6 | 12.4 | 4.9 | **0.04** |
| HPR | 8.3 | 8.7 | 9.3 | 4.9 | **0.04** |

**Table 3: Same as Table 1 but for ERA5 annual mean daily 10-m wind magnitude (m s$^{-1}$).**

| Annual mean daily evaporation changes by clusters, 1979-2023, ERA5 only | | | | | |
|---|---|---|---|---|---|
| | 1979-predicted (mm day$^{-1}$) | 2023-predicted (mm day$^{-1}$) | change (mm day$^{-1}$ decade$^{-1}$) | change (%) | p-value |
| Total | 1.4 | 1.4 | 0.00 | -0.2 | 0.96 |
| ZFL | 1.6 | 1.7 | 0.01 | 3.6 | 0.53 |
| WAA | 0.2 | 0.1 | 0.00 | -13.4 | 0.79 |
| LPR | 0.6 | 0.6 | 0.00 | 1.7 | 0.86 |
| CAA | 3.1 | 3.2 | 0.02 | 3.2 | 0.40 |
| HPR | 0.9 | 1.0 | 0.02 | 7.8 | 0.29 |

**Table 4: Same as Table 1 but for ERA5 annual mean daily evaporation (mm day$^{-1}$).**

A bias in the ERA5 evaporation could be a consequence of the bulk parameterisation not capturing the subgrid scale physics of the ocean-atmosphere interface across the broader SO, and that these physical processes have been enhanced over the past 45 years. For example, widespread shallow convection has been observed to produce precipitation, cold downdrafts, and gusty winds at the ocean surface on the scale of kilometres, beyond the horizontal grid spacing of ERA5 (Lang et al., 2021; Lange et al., 2022; Alinejadtabrizi et al., 2024). Further, the broader SO is renowned for intense winds and waves that produce abundant sea spray, which can lead to enhanced surface latent and heat fluxes compared with standard bulk calculations (Song et al., 2022; Richter and Veron, 2016; Hartery et al., 2020; Lin et al., 2024). Understanding the processes that are causing the ERA5 bias in

precipitation will also inform the assessment of coupled climate models, which suffer from a similar bias (Purich et al., 2018) and are even underestimating storm track trends against ERA5 (Chemke et al., 2022).

This increase in precipitation has further implications for the freshwater flux over the high-latitude SO. The hypothesised increased evaporation, predominantly in the region north of the storm track, is consistent with the observed increase in surface salinity north of 45° S (Swart et al., 2018). Further poleward, consistent with the observed intensified precipitation at MAC, a surface freshening has been observed (Durack and Wijffels, 2010; Durack et al., 2012). Assuming the storm track region typically extends over more than half of the hemisphere

longitudinally, and across the 50-60° S latitude band (Hoskins and Hodges, 2005), this implies that it covers an area of more than $1.3{\times}10^7$ km$^2$. Ignoring any changes in evaporation within the storm track region and extrapolating, the observed MAC precipitation increase of 260 mm over this area represents an additional ~3,400 Gt freshwater flux into the high-latitude SO in 2023 compared to 1979. The ERA5 precipitation intensification represents an additional ~1,100 Gt freshwater flux over this same time, an underestimation of ~2,300 Gt. These

high-latitude freshwater fluxes are substantial: for context, current Antarctic mass loss estimates from the 1990s to the present are of order 200-500 Gt y$^{-1}$ (Swart et al., 2023). While intensified precipitation does not raise sea levels like Antarctic mass loss, increased freshwater fluxes modify SO stratification and circulation and have far-reaching climate impacts. Coincident with surface freshening, high-latitude SO surface cooling from ~1980-2010 (Dong et al., 2023) is consistent with increased stratification and a weaker entrainment of warmer subsurface

waters into the surface layer and is supported by modelling studies examining the impact of SO surface freshening (Pauling et al., 2016; Purich et al., 2018). Furthermore, emerging evidence suggests the high latitude SO cooling is linked to the cooling in the eastern equatorial Pacific (Kang et al., 2023; Kang et al., 2023; Watanabe et al., 2024), highlighting the global consequences of increased precipitation across the SO storm track, as observed at MAC.

While MAC offers precious long-term observations over the SO, its single-point nature introduces potential scale mismatch with the nearest ERA5 grid-cell mean (0.25° × 0.25°, ~15-30 km), which may contribute to the observed biases in precipitation trends. Moreover, the extent to which MAC observations are representative of the broader SO area remains uncertain. Employing higher-resolution models or satellite products, as well as validating results

against other stations within this region, would help mitigate these limitations in future studies.

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

**Code and Data Availability**

The Macquarie Island observational data are available from the Australian Bureau of Meteorology (A-BoM) at http://www.bom.gov.au/climate/data/ and from the NCEI-NOAA Global Hourly Integrated Surface Database (ISD) at https://www.ncei.noaa.gov/products/land-based-station/integrated-surface-database. ECMWF reanalysis data (ERA5) can be accessed via the Copernicus Climate Data Store at https://cds.climate.copernicus.eu/. Topographic data (SRTMGL1) are available from the NASA Land Processes Distributed Active Archive Centre (LP DAAC) at https://lpdaac.usgs.gov/products/srtmgl1v003/. The HYSPLIT model is available from the NOAA Air Resources Laboratory at https://www.ready.noaa.gov/HYSPLIT.php. The MeteoInfo software (used to convert ERA5 GRIB files into HYSPLIT ARL format) is available from http://www.meteothink.org/.

**Acknowledgements**

This research was supported by the Australian Research Council (ARC) Special Research Initiative Securing Antarctica's Environmental Future (SAEF) (SRI20010005) and the ARC Centre of Excellence for 21st Century Weather (CE230100012). We also acknowledge support from the National Computational Infrastructure (NCI), Australia. We are sincerely grateful to the Australian Antarctic Division (AAD) and the Australian Bureau of Meteorology (BoM) for kindly providing the observational data and the accompanying photograph. We further appreciate Prof. Tim Woollings, the two anonymous reviewers, and the editorial team for their constructive comments, insightful suggestions, and helpful support, all of which greatly improved this work.

**Author Contributions**

Zhaoyang Kong, Andrew Prata, and Ariaan Purich performed most of the data analysis and visualization. Steven Siems, Peter May, Yi Huang, and Ariaan Purich provided the foundational framework and contributed the main parts of the initial manuscript draft. All authors provided editorial feedback on the manuscript. All authors read and approved the final version.

**Competing Interests**

The authors declare no competing interests.

**Appendix**

| Annual mean daily precipitation intensity changes by clusters, 1979-2023, MAC obs. and ERA5 | | | | | | |
|---|---|---|---|---|---|---|
| | | 1979-predicted (mm day$^{-1}$) | 2023-predicted (mm day$^{-1}$) | change (mm day$^{-1}$ decade$^{-1}$) | change (%) | p-value |
| Total | MAC | 2.6 | 3.3 | 0.16 | 28.1 | **0.00** |
| | ERA5 | 2.8 | 3.0 | 0.05 | 8.5 | 0.05 |
| ZFL | MAC | 2.9 | 3.1 | 0.03 | 4.7 | 0.50 |
| | ERA5 | 2.9 | 2.7 | -0.04 | -5.6 | 0.30 |
| WAA | MAC | 3.4 | 4.6 | 0.26 | 33.6 | **0.00** |
| | ERA5 | 4.2 | 4.5 | 0.07 | 6.9 | 0.17 |
| LPR | MAC | 3.8 | 5.4 | 0.37 | 43.7 | **0.00** |
| | ERA5 | 3.7 | 4.7 | 0.22 | 25.9 | **0.04** |
| CAA | MAC | 2.0 | 2.7 | 0.15 | 33.3 | **0.00** |
| | ERA5 | 2.3 | 2.5 | 0.04 | 7.2 | 0.24 |
| HPR | MAC | 0.8 | 1.1 | 0.05 | 28.6 | 0.08 |
| | ERA5 | 1.0 | 1.0 | 0.00 | -1.1 | 0.94 |

**Table A1: Numerical details extracted from Figure 5. (p < 0.05 indicating a statistically significant trend, shown in bold)**

| Annual accumulated precipitation changes by clusters, 1979-2023, MAC obs. and ERA5 | | | | | |
|---|---|---|---|---|---|
| | | 1979-predicted (mm) | 2023-predicted (mm) | change (mm decade$^{-1}$) | change (%) | p-value |
| Total | MAC | 933 | 1193 | 59.0 | 27.8 | **0.00** |
| | ERA5 | 1013 | 1098 | 19.2 | 8.3 | 0.06 |
| ZFL | MAC | 270 | 287 | 3.8 | 6.2 | 0.48 |
| | ERA5 | 268 | 255 | -2.8 | -4.7 | 0.47 |
| WAA | MAC | 212 | 355 | 32.7 | 67.9 | **0.00** |
| | ERA5 | 262 | 348 | 19.6 | 33.0 | **0.00** |
| LPR | MAC | 225 | 275 | 11.3 | 22.1 | 0.13 |
| | ERA5 | 223 | 237 | 3.2 | 6.4 | 0.58 |
| CAA | MAC | 175 | 212 | 8.4 | 21.1 | **0.03** |
| | ERA5 | 200 | 199 | -0.2 | -0.4 | 0.97 |
| HPR | MAC | 51 | 64 | 2.8 | 24.4 | 0.06 |
| | ERA5 | 61 | 58 | -0.7 | -4.8 | 0.67 |

**Table A2: Numerical details extracted from Figure 6. (p < 0.05 indicating a statistically significant trend, shown in bold)**

625