# Peer review of "Intensifying precipitation over the Southern Ocean challenges reanalysis-based climate estimates — Insights from Macquarie Island's 45-year record"

_EGUsphere, 2025_

## Referee Comment (RC1)

**Title: Intensifying precipitation over the Southern Ocean challenges reanalysis-based climate estimates – Insights from Macquarie Island's 45-year record**
**Author(s): Zhaoyang Kong et al.**
**MS No.: egusphere-2025-3496**
**MS type: Research article**

General Comments:

The paper is scientifically sound and the results are promising. However, there are a few items need to be addressed (see Specific Comments section). Therefore, my suggestion it is required some minor revisions before it can be accepted for publication.

Specific Comments:

1. ERA5 precipitation data and trend analysis

Because the size of Macquarie Island is very small, any analysis from 0.25x0.25 ERA5 data has a great uncertainty. This requires some discussions in paper. Also, ERA5 has an ERA5-Land hourly dataset which is on 0.1x0.1 grid. The model physics of land and ocean in this dataset should be represented better for Macquarie Island. I would like authors add one more line from 0.1x0.1 land dataset on Figure 1d and add some discussions. This line can replace 3-other-pts averaged line.

2. Section 2.2.5

I don't see the importance of LHF section in this paper. ERA5 already has precipitation, evaporation, and runoff variables. Either remove all the LHF discussions/plots, or explain it more clearly the role of LHF in this study.

3. Figure 1d

The values for trends are missing, I assume they should be displayed similar to Figure 5 (mm per decade).

4. L88, L102

Time, I suggest rewrite those two sentences using L112 format xx UTC (xx LST).

5. L158 and Fig 4b-middle

This sentence seems out of place. Either reorder figures or put this sentence in proper place. Also, it seems to me that it is redundant to plot LHF and evaporation on the same plot.

6. L168 3.1

I suggest put a abbreviation for each regime, similar to Fig. 2.

Technical Corrections:

2. L156, Eq (2)

$LHF = Lv/86400 \approx$ ....

3. Eq (2) and (3)

Change "," to "." at the end, or remove them.

4. Code and Data Availability

Add HYSPLIT model code links

---

## Referee Comment (RC2)

**Intensifying precipitation over the Southern Ocean challenges reanalysis-based climate estimates – Insights from Macquarie Island's 45-year record**

September 15, 2025

**General Comments**

This paper presents an interesting study on the discrepancy between the precipitation observations over Macquarie Island and the ERA5 reanalysis based climate estimate. They link this discrepancy to the limited representation of atmospheric moisture transport and increasing evaporation in ERA5.

The authors identify five different synoptic regimes for characterising the progression of mid-latitude cyclones and associated fronts along the SO storm track: zonal flow, warm air advection, low pressure, cold air advection and high pressure. Among these, only the warm air advection regime has a statistically significant trend in frequency of occurrence over the 45-year period (1979-2023), increasing from 63.2 days per year in 1979 to 78.5 days in 2023.

Over the same period, the intensity of the daily MAC precipitation increases 28.1% (2.6 to 3.3 mm day-1), while ERA5 only increases 8.5%, both statistically significant increases. The authors decomposed the increase in accumulated precipitation into the components arising from changes in frequency of occurrence and changes in mean daily intensity. They found that the shift in frequencies over the different regimes has a small impact on the overall increase in accumulated precipitation for both MAC and ERA5. Instead, the results suggest that it is the increase in the mean daily intensity of the different synoptic patterns that drives the increase in accumulated MAC precipitation.

Overall, the paper is very well written, and the results are clearly presented. Before recommending publication, I would like the authors to address some comments.

**1 Representativeness error**

The findings of this analysis are indeed interesting and very useful, but I believe the authors should emphasize the limitations of this work due to the usage of a single rain gauge. A single point measurement might not be representative of the ERA5 output which represents an average over a grid cell with a resolution of roughly 31km. The differences observed could stem from ERA5 model bias but also from mismatch bias (point vs cell-mean). A discussion on representativeness errors should be included.

**2 Risk of over-interpreting the results**

Considering that the study is based on a single measurement site, the authors should be very careful about drawing conclusions that are too broad. The discussion on the changes across the full SO track, starting line 335 feels a little bit too broad. This study is a step towards evaluation and understanding the changes across the full SO storm track, but a much broader evaluation will be required to fully understand the long-term dynamics in this area.

In the abstract,

*This precipitation discrepancy reveals moisture and energy budget biases in reanalysis over the SO ...*

In my opinion, there is not enough data to support this, and it's not clear how representativeness errors affect this study. The authors should use a more suggestive tone rather than a definitive statement.

**3 Line 326. ERA5 captures these trends in the synoptic meteorology, as MAC soundings are assimilated into the reanalysis and are heavily weighted given the remoteness of the location.**

This is a little bit misleading, since this statement seems to imply that it is only due to the MAC readings that ERA5 captures these trends. MAC soundings are surely important, but ERA5 heavily relies on satellite data also. Therefore, the authors should rewrite this phrase in a more suggestive tone.

**4 Line 266. For ERA5, only the trend in the daily intensity of the low pressure regime is statistically significant. Notably, the trend is negative for both zonal flow and high pressure regimes.**

The trends for zonal flow and high pressure regimes are not statistically significant. Therefore, the authors should not use the word *notably*. It could be argued that this should not be mentioned at all (trend being negative). The same applies for line 253

*This increase in warm air advection days largely comes at the expense of low pressure days, which decrease from 60.9 to 50.5 days, followed by cold air advection days, which decrease from 85.4 to 80.4*

The increase in warm advection is statistically significant, while the decrease in low pressure and cold air advection regimes is not. We lack statistical support to state that the increase in warm air advection comes at the expense of the other two regimes.

**5 IMERG**

Have the authors considered exploring different global precipitation products, such as IMERG, to check whether similar differences in trends are observed? One potential

issue is that IMERG is calibrated using measurements from Macquarie Island, but an IMERG–ERA5 comparison would still be interesting for this region. I am not suggesting that the authors undertake this analysis for the present paper, but a discussion on the topic could be interesting.

---

## Author Comment (AC1)

**Final response to the reviewers' comments**

Date: October 20th, 2025

Title: Intensifying precipitation over the Southern Ocean challenges reanalysis-based climate estimates – Insights from Macquarie Island's 45-year record

MS No.: egusphere-2025-3496

Correspondence to: Zhaoyang Kong (zhaoyang.kong@monash.edu)

First of all, we would like to express our sincere gratitude to the editorial team for their assistance and to the two reviewers for their constructive suggestions, which have significantly improved the content of this paper. We provide a point-by-point response to all comments from the reviewers in the following pages. For convenience, we abbreviate the line numbers as "L". Please note that the specific line numbers in our responses refer to the locations in the revised version, and all changes are highlighted in yellow. Before the details part, we provide the following summary of our responses:

**Reviewer #1's Comments:**

- 1. For Specific Comment 1 point 1, we have added a paragraph (L391-395) at the end of Section 4 to discuss the limitations. Regarding the second point, the ERA5-Land reanalysis product does not include data for Macquarie Island (MAC) and its surrounding regions (all values are NaN, as shown in Rev1.Figure 1). Therefore, we are unable to conduct further data analysis.
- 2. For Specific Comments 2 and 5, both comments discussed about the role of latent heat flux (LHF) in this work. Following the reviewer's suggestions, we have simplified some of the relevant lines, but we still hope to retain the LHF descriptions and plot in the paper to quantify the impact of the precipitation on the energy budget.
- 3. All other points have been revised according to the reviewer's suggestions and are highlighted in yellow in the new version.

**Reviewer #2's Comments:**

- 1. For Comment 1, similar to Reviewer #1's Specific Comment 1, we have added the paragraph (L391-395) at the end of Section 4 to describe the limitations.
- 2. For Comments 2-4, we have rewritten each point based on the reviewer's suggestions and included them in the new version.
- 3. For Comment 5, regarding the comparison between MAC observed precipitation records and the Integrated Multi-satellite Retrievals for GPM

(IMERG) satellite product, this is part of our ongoing work, and we are looking forward to sharing the results with readers in the near future.

In addition to the comments from the reviewers, we have made five minor revisions in the original submission:

- 1. We have revised the reference annual (or decadal) average growth rates in Figures 5 and 6 (and the appendix tables), as well as Tables 3 and 4. The correct reference period should be the interval from 1979 to 2023 (44 years), rather than including all years (45 years). This adjustment corrects an underestimation of ~2.3% (1/44) compared to the previous version. Please note that all the modifications are provided only as reference values in the figures or tables, and are not mentioned in the text.
- 2. L140: " $f_{i,j}$  represents frequency (number of days)" should be corrected to " $f_{i,j}$  represents frequency proportion in percentage" (as highlighted in yellow).
- 3. L20: We have added "..., consistent with a poleward shift in the storm track" (as highlighted in yellow).
- 4. L89-95: The original sentence, "However, the dataset includes values with a 0.1 mm resolution (Tansey et al., 2022) in recent years, suggesting the use of higher-resolution sensors during this period" was not sufficiently accurate. We revised this description in conjunction with Reviewer #1's Specific Comment 4 (as highlighted in yellow). In addition, we removed the citation to Tansey et al. (2022) here and made the corresponding change in the reference list.
- 5. We have revised the citation description in L118 to make it a complete sentence (as highlighted in yellow), while keeping the two cited references unchanged.

We sincerely apologise for these errors/revisions.

Moreover, we have added lines in the Acknowledgements section to thank Prof. Tim Woollings, the two reviewers, and the editorial team. At the co-authors' request, we have included their respective middle names in the author list (L4-5).

Below are our detailed responses to **Reviewer #1**'s comments. Once again, we would like to express our gratitude to the reviewer. For convenience, the reviewer's original comments are in a grey background.

**Specific Comments:**

**1. ERA5 precipitation data and trend analysis**

Because the size of Macquarie Island is very small, any analysis from 0.25x0.25 ERA5 data has a great uncertainty. This requires some discussions in paper.

Also, ERA5 has an ERA5-Land hourly dataset which is on 0.1x0.1 grid. The model physics of land and ocean in this dataset should be represented better for Macquarie Island. I would like authors add one more line from 0.1x0.1 land dataset on Figure 1d and add some discussions. This line can replace 3-other-pts averaged line.

**Reply:**

Thank you for this suggestion. Regarding the first point, we have added a description of this limitation in the last paragraph of Section 4 of the revised version from L391-395 (yellow highlighted). Regarding the second point, unfortunately, the "Total Precipitation" data for Macquarie Island and its adjacent area in the ERA5-Land dataset are all in NaN, meaning that ERA5-Land does not take this island into account. Please see Rev1.Figure 1 for an example at 08:00 UTC on February 20, 2019.

Rev1.Figure 1: Coverage of the ERA5-Land dataset total precipitation variable for (a) the global area and (b) Macquarie Island and its adjacent area (54-55°S, 158.5-159.5°E), an example at 08 UTC on February 20, 2019, with NaN (in grey) and non-NaN (in light blue).

**2. Section 2.2.5**

I don't see the importance of LHF section in this paper. ERA5 already has precipitation, evaporation, and runoff variables. Either remove all the LHF discussions/plots, or explain it more clearly the role of LHF in this study.

**Reply:**

We apologise for not making the role of latent heat flux (LHF) clear. Given that precipitation/evaporation serves as a constraint on the surface energy budget (L38-39), in this paper, we hope to provide an initial estimation for the energy budget bias in ERA5 over the Southern Ocean region represented by MAC through LHF, based on the bias of precipitation trends (L349-350). And the purpose of Section 2.2.5 is primarily to provide an intermediate step for this rough estimation of LHF (bias) from precipitation trend (bias), as referenced in L61 and L350. Although the description of both LHF and evaporation may be redundant, it may assist in providing a quick understanding for potential readers from other fields. Therefore, we hope to retain Section 2.2.5 on LHF, and we have streamlined it, as seen in L158-160.

**3. Figure 1d**

The values for trends are missing, I assume they should be displayed similar to Figure 5 (mm per decade).

**Reply:**

Thank you for this comment. Yes, because trends of MAC observation and ERA5 nearest in both Figure 1d and Figure 6a share the same data, we omitted it to avoid duplication. However, we have added values for trends (mm per decade) to Figure 1d in the revised version.

**4. L88, L102**

Time, I suggest rewrite those two sentences using L112 format xx UTC (xx LST).

**Reply:**

Thank you. We have made changes in these two places to include UTC or LST, and we now consistently use the 24-hour format, expressed as HH:MM. Please see the yellow highlights in the new version (corresponding to L89-92 and L104-106). Additionally, the formats for LST and UTC at L115 and L131 in the new version have also been modified.

**5. L158 and Fig 4b-middle**

This sentence seems out of place. Either reorder figures or put this sentence in proper place. Also, it seems to me that it is redundant to plot LHF and evaporation on the same plot.

**Reply:**

We appreciate the reviewer's suggestion. We have streamlined the paragraph from the old version at L158 into the new version at L160, this point is aiming to confirm that ERA5 used the coefficient of 28.94 for the evaporation to LHF conversion.

Same with the reply of Comment 2, if possible, we would like to retain both LHF and evaporation in Fig. 4b-middle. While this might appear redundant, considering that some readers from other fields may find it difficult to immediately relate precipitation/evaporation to LHF, it could be helpful for them. We hope this clarification addresses the reviewer's concern.

**6. L168 3.1**

I suggest put an abbreviation for each regime, similar to Fig. 2.

**Reply:**

Thank you for the reminder. We have added abbreviations for each regime. Please see the yellow highlights in the new version (corresponding to L173-188).

**Technical Corrections:**

**2. L156, Eq (2)**

 $LHF = Lv/86400 \approx ...$

**Reply:**

Eq. (2) is the ratio of LHF to evaporation. We have modified Eq. (2) and its related description. Please see the highlighted parts in L158-160.

**3. Eq (2) and (3)**

Change "," to "." at the end, or remove them.

**Reply:**

We appreciate the reviewer for pointing out the errors. In the new version, we have adjusted the sentences for both Eq. (2) and Eq. (3) (L159-160 and L165-166).

**4. Code and Data Availability**

Add HYSPLIT model code links

**Reply:**

We thank the reviewer for this comment. We have added the links to the HYSPLIT model and MeteoInfo software in the Code and Data Availability section.

Below are our detailed responses to **Reviewer #2**'s comments. We would like to express our gratitude to the reviewer again. And please note that the reviewer's original comments are in a grey background.

**1 Representativeness error**

The findings of this analysis are indeed interesting and very useful, but I believe the authors should emphasize the limitations of this work due to the usage of a single rain gauge. A single point measurement might not be representative of the ERA5 output which represents an average over a grid cell with a resolution of roughly 31km. The differences observed could stem from ERA5 model bias but also from mismatch bias (point vs cell-mean). A discussion on representativeness errors should be included.

**Reply:**

Thank you for this suggestion. We have added a paragraph at the end of Section 4 discussing the limitations of the MAC observation site and its potential scale mismatch with ERA5, as the reanalysis grid-point data represent area-averaged values at their given resolution. (Please see the highlighted part in L391-395.)

**2 Risk of over-interpreting the results**

Considering that the study is based on a single measurement site, the authors should be very careful about drawing conclusions that are too broad. The discussion on the changes across the full SO track, starting line 335 feels a little bit too broad. This study is a step towards evaluation and understanding the changes across the full SO storm track, but a much broader evaluation will be required to fully understand the long-term dynamics in this area.

**In the abstract,**

This precipitation discrepancy reveals moisture and energy budget biases in reanalysis over the SO ...

In my opinion, there is not enough data to support this, and it's not clear how representativeness errors affect this study. The authors should use a more suggestive tone rather than a definitive statement.

**Reply:**

Thank you. Based on your suggestion, we have updated the rewrite of L335 and the corresponding part in the Abstract, which can now be found in L337-339 and L23-25 in the new version.

3 Line 326. ERA5 captures these trends in the synoptic meteorology, as MAC soundings are assimilated into the reanalysis and are heavily weighted given the remoteness of the location.

This is a little bit misleading, since this statement seems to imply that it is only due to the MAC readings that ERA5 captures these trends. MAC soundings are surely important, but ERA5 heavily relies on satellite data also. Therefore, the authors should rewrite this phrase in a more suggestive tone.

**Reply:**

Thank you for your suggestion. The text originally located at L326 has been revised and now corresponds to L327-329 in the updated version.

**4 Line 266. For ERA5, only the trend in the daily intensity of the low pressure regime is statistically significant. Notably, the trend is negative for both zonal flow and high pressure regimes.**

The trends for zonal flow and high pressure regimes are not statistically significant. Therefore, the authors should not use the word *notably*. It could be argued that this should not be mentioned at all (trend being negative). The same applies for line 253

This increase in warm air advection days largely comes at the expense of low pressure days, which decrease from 60.9 to 50.5 days, followed by cold air advection days, which decrease from 85.4 to 80.4

The increase in warm advection is statistically significant, while the decrease in low pressure and cold air advection regimes is not. We lack statistical support to state that the increase in warm air advection comes at the expense of the other two regimes.

**Reply:**

Thank you. Following your suggestion, we have rewritten the sentences in the old version at L266 and L253. Please see the highlighted parts in the new version at L266-268 and L253-254.

**5 IMERG**

Have the authors considered exploring different global precipitation products, such as IMERG, to check whether similar differences in trends are observed? One potential issue is that IMERG is calibrated using measurements from Macquarie Island, but an IMERG–ERA5 comparison would still be interesting for this region. I am not suggesting that the authors undertake this analysis for the present paper, but a discussion on the topic could be interesting.

**Reply:**

Thank you for this comment. Yes, the comparison between MAC observations, ERA5, and GPM-IMERG is part of our ongoing work. We are very much looking forward to presenting the results in the near future.

Finally, we would like to sincerely thank the two reviewers again for their valuable suggestions and the efforts they made to improve this work.